

# Good Vibrations: Living with the Motions of our Unsettled Planet

Tamsin Badcoe[1], Ophelia Ann George[2], Lucy Donkin[3], Shirley Pegna[4], John Michael Kendall[2,5]

[1]Department of English, University of Bristol, Bristol, UK
[2]School of Earth Sciences, University of Bristol, Bristol, UK
[3]Department of History of Art, University of Bristol, Bristol, UK
[4]Sound Artist, Bristol, UK
[5]Now at Department of Earth Sciences, University of Oxford, Oxford, UK

*Correspondence to*: ophelia.george@bristol.ac.uk

**Abstract.** Historical commentary in the aftermath of large earthquakes has frequently noted unscheduled ringing of church
bells excited by the shaking around them. These purported unscheduled bell ringing events were caused not only by near earthquakes but also by distant incidents. To investigate this phenomenon, as part of the Brigstow Institute funded Unsettled Planet Project, we installed a state-of-the-art broadband seismometer in the Wills Memorial Building tower to record how Great George (the tower bell) responds to the restless world around him. The installed seismometer has been recording activity around and within the tower on a near continuous basis since 23 March 2018. Here, we present the signals recorded by the
seismometer as Great George overlooks the hustle and bustle of the city around him and investigate how connected we are to our "unsettled planet", even from our tectonically quiet setting in Bristol. We find that the seismometer not only shows the hem and haw of activity in and around Bristol, but also brings to light earthquakes from as nearby as Lincolnshire, UK, or as far away as Fiji ~ halfway around the world. In order to contextualise our findings, our project also considers what determines how people have responded to earth shaking events, drawing on both historical and recent examples, and looks to contemporary
art practice in order to consider how an awareness of our "unsettled planet" can be communicated in new ways.

## 1 Introduction – the Unsettled Planet Project

People live with the ground shaking on a daily basis but when and why we become aware of this is contextual. To varying degrees our capacity to dwell on the surface of the Earth is shaped by a learned tolerance of vibrations – sometimes benevolent and commonplace, at other times potentially calamitous – beneath and through our spaces of habitation. From the measurable
footfall of humans walking along a pedestrian street in Bristol and the locomotion of individual elephants in Kenya, to the co-ordinated motion of a stadium crowd and the combined impact of traffic noise, our world reverberates with the forces generated by human, animal and vehicular activity. The resulting vibrations go largely unnoticed as the ground motion that they produce is relatively small and fades into the hustle of everyday life. More attention grabbing, of course, are the vibrations produced by phenomena such as earthquakes.



Whether large or small, these vibrations provide vital information not only about the settings in which we live but also about the natural world around us and how we react to it. Just as footsteps are easier to discern when made on a creaky floor than they are on solid ground, so too does the setting in which Earth-shaking occurs affect our perception of events. The 1985 magnitude 8 earthquake, 350 km away from Mexico City, for example, caused significant damage and loss of life. The extent

of the damage was due to the nature of the ground beneath the city (Seed et al., 1988). Centuries earlier, as Nicols (2014) observes of the devastating 1755 Lisbon earthquake, "the historic Baixia area on the north side of the Tagus, the seat of government, with narrow streets and timber-built houses, rested on water saturated with alluvial sediment" that "liquefied during the earthquake and lost its bearing strength during the shockwaves. Down the coast, Tavir, sitting on limestone met with few casualties". Furthermore, human-induced earthquakes in Lancashire – due to hydraulic fracture stimulation (fracking)

– register differently in the public consciousness (Williams et al., 2017) than much larger natural earthquakes in more seismically active regions such as Japan. An earthquake can thus be a disaster for one person, slightly worrying for another and barely perceptible for many more. The factors shaping individual responses might be simply determined by proximity to an earthquake but are more often influenced by other considerations including the underlying geology, local culture, collective memory, a familiarity with the associated phenomena, and the inequalities of disaster management. Living on an unsettled

planet such as ours is tolerable for most of the time, but at others can be catastrophic.

This paper offers reflections on an interdisciplinary project that explored the relationship between scales of Earth-shaking and thresholds of habituation. A response to the theme of "living well with uncertainty" proposed by the University of Bristol's Brigstow Institute, the project brought together a sound artist and researchers from the Arts, Humanities and Earth Sciences at

the University. We found ourselves particularly interested in how people react to first-hand experiences with the fact that the ground is not stable or firm. At one extreme, this involved thinking about living in earthquake prone regions, and at the other, considering how someone from the countryside might find it unsettling to live in a city with heavy traffic vibrations. Indeed, as we discovered, thinking about how people live with the ground shaking on a daily basis does not just apply to distant places, rich in seismic activity, poised on fault line, but can also be used to consider our own located positions, which for the purposes

of the project, took in the steep incline of Park Street and the Wills Memorial Building in Bristol: surely, the epitome of solid ground. Even familiar territory, we found, is constantly on the move, impacted both by our own bodies and by seismic events across the world. In the School of Earth Sciences, for example, our conversations took place against a background of heavy traffic vibration caused by buses passing by immediately outside the windows of the Wills Memorial Building: vibrations ultimately equivalent to those that might reach Bristol from a distant earthquake. In combining approaches drawn from

geophysics, the humanities, and the arts, then, we connected different categories of knowledge-making in order to grasp what is there or what could be there, not only under, but also passing through, the ground beneath our feet, and made links between what we know and what we imagine. The project focused on the current discourses surrounding controversial issues such as fracking but looked at how past perceptions of seismic and volcanic hazards have evolved over time. We specifically aimed to probe the effects of seismic vibrations from regions of the Earth that are both near to and far from our relative haven in





Bristol and to reflect on how those events are perceived by those around us. Our group aim was to engage with the sounding phenomena around us, thus enabling us to perceive the world in a different, and more connected way.

Within our interdisciplinary team of co-authors, we found marked differences in our ways of working. It was easy to recognise this methodological range, but its implications were harder to fathom. We could use labels that corresponded to the badges of

our professions – Earth Sciences, literary and historical studies, artistic practice – but the project asked us to think about our default settings and manner of approaching things. Some of us are more attentive to sounds, some of us to words, some to images, some to things, and some to numbers. Some of us think mainly in decades and centuries, others in minutes and in millions of years. We did not strain to find common ground; after all, we have a collective interest in the ground beneath our feet. It was always easy to talk, but this has involved a particular kind of conversion, which one participant described as

"specialists speaking colloquially". Of course, we are also more than our disciplines. What we each brought to the discussion was dependent on the parts of the Earth's surface with which we are familiar: a frame of reference which came partly from professional activities but also from personal trajectories. Anecdotes, questions, lines from a poem, an image, identities; all of these provided points of connection, but we ultimately found collective stimulus in a bell.

## 1.1 Great George and the Wills Memorial Building

Great George (the bell in the tower of the Wills Memorial Building) became something of an emblem for us, in that it offered a fusion of 'vibrant matter' and symbolic resonance (Bennett, 2010). As a material object specifically designed and fabricated to resound in a particular way, the bell participates physically in the marking of time but also acts as a reminder of how bells have historically sounded in times of celebration, admonition, and mourning. By using the bell as an instrument for registering vibration, whether that caused by the passage of buses on Park Street or that generated by seismic activity a thousand kilometres

away, we reflected on technologies of recording, both those designed with a specific purpose in mind and those repurposed and used inventively and creatively. In its capacity to register vibrations, then, the bell's purpose as percussive instrument was revised, using the addition of a seismometer (Fig. 1), to become an instrument of measurement: a transformation that also asked us to think about how solid bodies, including our own, register the unpredictable motion of our unsettled planet. Bells have often been anthropomorphised, given human names and marked with inscriptions couched in the first person. If the place

of Great George in the project lay partly in the rich potential of using a bell as an object to think with, its importance for us also owed something to the fact that it was physically present above us in the building in which we generally met, and audible where ever we were in the city. Great George is an object that can be visited, touched, and even swung: an object that was already multi-vocal and expansive, in terms of both its E-flat tone – audible up to 20 kilometres away – and its twitter feed, with a potentially global reach. The material qualities of the historic bell, cast in 1924, became particularly compelling when

put into dialogue with modern scientific equipment, which was as exotic and mysterious to some of us as it was familiar to others.



## 1.2 Thumbing through the Special Collections

The project also involved encounters with objects that were more familiar to those of us working in the arts and humanities. A workshop in the Special Collections of the University Library allowed us to look together at printed material on earthquakes ranging from the sixteenth to the nineteenth century, drawn mainly from the Eyles Collection (Eyles and Eyles, 1679 - 1983) Here, the objects of our attention were sometimes fragile books resting on foam cradles, pages held down with snake weights – works with evocative paragraph-long titles such as "A true and exact relation of the most dreadful earthquake which happened in the City of Naples and several parts of that Kingdom, June the 5th, 1688: whereby about forty cities and villages were either wholly ruin'd or extreamly damnified; eight thousand persons destroy'd and about eight hundred wounded; of which four hundred were digg'd out of the ruins, and many others miraculously preserved, translated from the Italian copy printed at Naples, by an eye-witness of those miserable ruins" (J.P., 1688). We scrutinized striking images such as Athanasius Kircher's cross-sections of the Earth (Kircher, 2015), previously met as disembodied images on the web (a very different experience from turning the pages of *Mundus Subterraneus* in its published book form). Voices from the past included descriptions of the sensations and emotions engendered by moving ground, but also attempts to explain seismic activity, whether in terms of human wrongdoing and divine punishment or in terms of natural process. For the scientists amongst us, this brought home not only the differences in how science was conducted in previous centuries but also the resonances still perceptible with ideas held today. For those of us accustomed to the objects and setting of a rare books collection, it was appreciable how the process of collaborative looking and reading differed from the sustained attention and focused questions of personal research. As we all circulated amongst the works, pointing out aspects and reading out passages – some of us more drawn to human testimony, others to scientific thinking – we provided contexts and explanations drawn from our own disciplinary frameworks.

## 1.3 Public colloquium

A one-day public colloquium allowed us to share some of our developing ideas with a wider audience and to benefit from the expertise and insights of others. Part of the day adopted a traditional format of short papers. Three invited speakers discussed projects that independently involved dialogue between disciplines and audiences. Stephen Vaughan, a photographer from the University of Bath Spa, spoke about his work on the impact of earthquakes in Japan and the United States. His focus on earthquake-induced tsunamis encouraged us to include water more fully in our thinking. By exploring how individual seismic events are discernible in the geological and dendrochronological record on both shores of the Pacific, he also gave us another way to think about distance and how seismic activity connects places. Paula Koelemeijer, an earth scientist then at the University of Oxford, reported on research conducted with Beth Mortimer (Mortimer et al., 2018), a biological scientist at the University of Bristol, bringing animals into the conversation. Not only was this research team able to track the movement of the elephants, differentiate between them and determine the kind of ground they were traversing, but they hypothesized that the elephants themselves were possibly able to communicate in this way, picking up on different frequencies produced by the tread of fellow elephants across distances of 10s to 100s of kilometres. Paul Denton from the British Geological Survey spoke



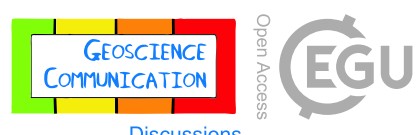

about crowd generated earthquakes. Since the bulk of conversations about human-induced seismicity surrounds activities such
as fracking, it was surprising for many to learn that amassed human footfall during outdoor rock concerts or football games
could result in ground vibrations equivalent to those induced by a magnitude 2-3 earthquake a few kilometres away. Low cost
seismometers, built from Lego, for example, and installed in schools can pick up such vibrations as well as other earthquakes,
extending the kinds of possible audience invited to attend to the movement of the unsettled planet.

Within the lecture theatre, charismatic objects from science and artistic practice helped to make ideas concrete. Speakers passed
around 3D printed models of the Earth and a fired-clay disc inscribed with a sound wave in the manner of an LP. At the edge
of the stage, a different kind of speaker – a huge amplifier – formed an intriguing presence, already reminding us of the sonic
quality of seismic activity before allowing everyone to hear and feel the movement recorded by Great George the bell.
Participants also gained a new awareness of the unsettled ground beneath the building by inhabiting other spaces, heading up
the tower to meet Great George himself and look out over his soundscape, and sitting in a van at the base of the tower to listen
to the amplified steps of passers-by (Fig. 2).

These activities were carried out concurrent with our seismic instrumentation of the bell tower. In the following sections, we
describe the setup of our experiment as well as the data that we gathered from the instrument between March 2018 and January
2019. We describe the data gathered in the context of how our bell tower is positioned not only within the interactions of our
city but also in a larger global picture.

**2 Data Summary – monitoring the world from a bell tower**

To quantify the contemporary vibrations of our unsettled planet, we used two methods to collect seismic data. The bulk of
these data were collected in and around the Wills Memorial Building (WMB). The WMB tower is a Neo Gothic reinforced
concrete structure which stands 65.5 meters above the street level and is 16 square meters at its base. Topping the tower is an
octagonal lantern which houses Great George. Great George is a large E-flat bell that measures 2 meters tall by 2.5 meters
wide and weighs approximately 9.5 tonnes (Ringing For England, 2019; Towerbells.org, 2019), making it one of the largest
bells in the UK. Great George rings hourly by the action of an external clapper except on special occasions such as the
University Charter day when it is rung in the traditional way by experienced bell ringers using an internal clapper.

The first of the two data collection methods utilized a 10 Hz vertical geophone which was embedded in the shallow subsurface
outside the WMB (Fig. 2). A geophone is a sensitive instrument that converts ground motion into an electrical signal via a
magnet suspended within a coil on small springs. The data collected by the geophone, such as the footsteps of pedestrians on
the pavement, were then converted into sound and played back through an audio system in real time. This allowed workshop
participants to have both a visual and audio illustration of the vibrations recorded by the geophone.



The second method of data collection involved the installation of a Nanometrics Trillium 120 PA 3-component broadband seismometer, connected to a Taurus datalogger in the WMB tower (Fig. 1), to monitor the response of Great George not only to the activity in and around the building but also to signals from much farther distances. Such instruments are routinely

deployed at seismic stations around the world and are used to monitor the Earth's seismic activity. This station was given the name GT01 and will be referred to as such in the figures that follow. A broadband seismometer allows for the accurate recording of seismic data (vibrations) in a broad range of frequencies, which is ideal for capturing the variety of signals reaching the tower. The 3 components of the seismometer record ground velocity in 3 orthogonal directions: East-West (HHE), North-South (HHN) and up/down or vertical (HHZ). The seismometer collected data at a rate of 100 Hz (100 samples per

second) which effectively captures signals with frequencies up to 50 Hz. At the low end, the instrument is sensitive to frequencies less than 0.01 Hz, or greater than 100 seconds in a period. These data were recorded intermittently between 23 March 2018 and January 2019 and several interesting signals have been extracted from the data including the magnitude 4.2 Lincolnshire, UK earthquake that occurred on 9 June 2018. The seismometer also recorded many distant – so-called teleseismic – earthquakes that occurred many thousands of kilometres away from the UK. In addition, several other noteworthy

observations have been made. These include differences in signal recording between night/day and weekday/weekend, variations in signal amplitude measured on the vertical and horizontal channels, and differences between the recordings made on the seismometer in the tower and other seismometers that were running on a lower floor of the Wills Memorial Building.

## 2.1 Signals from the bell ringing

On most days, Great George rings hourly by the action of an external clapper striking the exterior of the bell. The sound

produced by this ringing can be heard as far as 1.6 km (1 mile) away; the maximum distance at which the bell can be heard extends up to 20 km (12 miles) (Ringing For England, 2019) when the internal clapper is used on special occasions. The seismic signal produced by the ringing of the bell stands out from the background activity around the tower because it is produced by a source that is near the tower seismometer. The signal also contains a distinctive set of frequencies, which gives the bell its recognisable sound.


There are a number of ways of visualizing seismic data; commonly, signals are displayed in the time domain where changes in amplitude of the waves are shown as a function of time (e.g. Fig. 3). Another method of visualizing seismic data is to display the signal in the frequency domain – here, the amount of energy in a frequency band is shown as a function of time. Such data are typically presented in a spectrogram as is done in Fig. 4, which represents the energy in each frequency band as a colour;

warmer colours indicate stronger energy.

Figure 3 shows a day of data recordings on all 3 components of the seismometer. These recordings were made on 23 May 2018, the University Charter Day when the bell was rung by a group of bell ringers (see time lapse video of the bell and Charter





day bell ringing: https://vimeo.com/317784573 accessed: 08/04/2020). The regular bell chimes are less than half the amplitude

of the Charter Day bell ringing which occurs at roughly 12:50 PM; the internal clapper produces a much louder sound than the external clapper. A zoomed in view of one of the bell chimes (Fig. 4) shows that the individual chimes are clearly visible in the data; there is also a low amplitude signal, most visible before the first chime, created by the grinding motion of the external clapper leaving its housing to strike the bell.

## 2.2 Variations in activity levels near the tower as a function of time

In addition to the signals produced by the ringing bell, there are three noteworthy features that are striking when looking at a full day of activity near the tower. First, the station is always recording activity – the area is never quiet. The tower sits along one of the major thoroughfares into and out of Bristol city centre, including an often-used route to and from the Bristol Royal Infirmary (hospital). Second, the data show a marked difference in activity near the tower during the day as opposed to the night-time. And third, there are several more hours of overnight activity on the weekends compared to a typical weeknight.

Figure 5 shows two days of activity, presented as one hour of data in the time domain per line (much like an old fashioned helicorder). The data for Thursday (14 June 2018, 00:00 UTC to 15 June 2018 00:00 UTC) (Fig. 5a) shows that there are fewer bursts of activity in the early morning hours than there are later in the day stating at about 5 AM UTC until ~ 22:00 UTC. Fig. 5b shows activity recorded from 12 AM UTC on 16 June to 12 AM UTC of 17 June, which was a Saturday. There is more activity late into the Saturday night and early in the morning following Friday night.


Figure 6 displays the same data presented in Fig. 5 as a spectrogram. Again, both figures show higher amounts of energy/vibrations recorded during the daytime hours starting at about 6 AM and continuing throughout the day until about 10 PM at night on a typical weekday. They also show that this period of heightened activity extends almost 24 hrs on the weekend. This is particularly clear on the condensed seismogram displayed above the two spectrograms. These examples confirm what

we already know, namely that there is more activity in a city centre during work hours and also late at night on the weekends; however, the recorded data also evidences the extent to which our cities are never silent, existing as environments continuously animated by ground vibrations. A notable departure from this constant hum of activity comes in the wake of the 2020 COVID-19 related lockdowns. The nearly global stay-at-home orders have resulted in a corresponding global decrease in seismic background noise levels. In Brussels, Belgium, for instance, the reduction in background noise is such that in some cases,

seismic events at the same high frequency levels as the background noise can be detected and surface stations have the same signal quality as borehole stations buried at depths of 100 metres (Gibney, 2020).

## 2.3 Regional and teleseismic earthquakes

During the ~1-year deployment of a seismic station in the WMB, several earthquakes were recorded from distant sources. Many of these occurred along plate boundaries, where tectonic plates collide, separate or slide past each other. Here we discuss

an event from Indonesia, in which strike-slip faulting and the ensuing tsunami resulted in significant damage. We also show





an event from Greece, where the plate boundary is more diffuse, but still leads to large earthquakes. Finally, we consider events from Hawaii, recorded during an eruptive sequence of Kilauea Volcano. Here, the mantle is upwelling in a plume-like structure, forming the chain of Hawaiian Islands. But first, we consider an earthquake much closer to home, which occurred on an ancient fault that is still settling down in response to the retreat of the large glacial sheets that once covered the British Isles.

Cumulatively, these signals document the voice of a restless planet.

When an earthquake occurs, it releases energy in the form of seismic waves which radiate away from the earthquake source in all directions. This energy can be picked up by seismometers because they are highly sensitive instruments capable of measuring vibrations not only in the immediate vicinity of the instrument, but also from regional and teleseismic distances.

What distinguishes a regional from a teleseismic event is a bit arbitrary, but generally, teleseismic events are considered to be those that occur at distances greater than 1000 km and regional events are closer. Charles Richter (1935) established an empirical magnitude scale for assessing the size of local seismic events (abbreviated ($M_L$), defined as those closer than 500 km from a seismic station. There are several magnitude scales in use today, but Richter's local magnitude scale is among the most commonly used. When describing the size of larger seismic events (magnitude 4 and above) it is more common to use

the moment magnitude scale (abbreviated $M_w$) which is based on an assessment of the work (force times distance) done during earthquake rupture (Hanks and Kanamori, 1979). Where possible, seismologists prefer to use the $M_w$ scale over the $M_L$ scale, as the former is directly related to physical processes (i.e., the slip on a fault area in rock of a given strength) (see, e.g., Kendall et al. (2019)).

An example of a local or near-regional earthquake was recorded by the seismometer on 9 June 2018. This magnitude 4.3 earthquake occurred near Grimbsy, in Lincolnshire, UK. Signals from regional (or teleseismic) earthquakes look very different at distant stations than they do at stations near the source (compare the images in Fig. 7). As a seismic wave propagates through the Earth, its signal gets weaker and weaker as the seismic wave spreads out – this is known as spherical divergence or geometrical spreading. Furthermore, the Earth acts as a natural filter, absorbing high frequency energy. Distinguishing these

signals from the background noise at distant stations often requires the application of a filter to remove cultural noise like that seen in Fig. 6 in the frequency range of 5-20 Hz. For most of the distant earthquakes recorded in the tower, a filter of 0.1 to 1 Hz was used, which is in a range well below the dominant cultural noise from traffic, people and machinery.

The seismic signal recorded near the Lincolnshire earthquake clearly shows the onset of the seismic wavetrain at station LMK,

which is near Market Rasen. Seismic signals from an earthquake are comprised of many so-called phases. The clearest distinction to make is that between the first arriving P-wave, which is a wave that propagates as a series of compressions and rarefactions, and the later arriving S-wave, which propagates as oscillations transverse to the direction of wave propagation. The P-wave and S-wave arrivals are clearly visible at station LMK, with the P-wave primarily visible on the vertical component of the seismometer (HHZ) and the S-wave visible on the two horizontal components (HHE and HHN). In contrast, the signal





from this earthquake recorded on GT01 in the WMB tower is much weaker in amplitude; the signal recorded on GT01 is about $1/10^{th}$ the amplitude of that recorded on station LMK. Furthermore, the onset of the P-wave and S-wave is much less clear.

The Lincolnshire event was relatively small compared to several other large and devastating earthquakes that occurred in 2018. As indicated above, Richter's original magnitude scale is only appropriate for local events, so instead, these large events are

reported in the moment magnitude ($M_w$) scale. One such event was the 28 September 2018 magnitude $M_w$ 7.5 Palu, Indonesia earthquake which resulted in significant loss of life and property. It was the deadliest earthquake of 2018, and the earthquake and resulting tsunami and mudflows led to an estimated death of greater than 4340 people, over 10,000 injuries and the destruction of over 70,000 homes (Goda et al., 2019).

Although this earthquake was almost 110° (over 12,000 km) away from Bristol, the seismometer was able to detect the earthquake over 14 minutes after it occurred (Fig. 8). What was a national disaster and a source of human loss in a developing nation, was only perceptible on sensitive seismic instruments in the UK. More poignantly, an equivalent sized earthquake in Japan would likely not lead to any loss of life. Whilst Japan and Indonesia experience large earthquakes every year, as they lie on the boundaries between tectonic plates, Japan has the money to engineer and construct earthquake resilient buildings and

infrastructure.

When examining data from distant, large earthquakes, early researchers (Oldham, 1906; Mohorovičić, 1910; Jeffreys, 1926; Lehmann, 1936) could identify simple arrivals from P and S-waves, but they could also see other arrivals as seismic energy propagated and reverberated within the Earth. Some of these additional phases (Pdiff, SP, PKKP, etc.) can be seen in Fig. 8

and Fig. 9. The first signal to arrive at GT01 is the P-wave (Pdiff) that transits the Earth's mantle and diffracts along the core-mantle boundary, which lies nearly 3000 km below our feet. Others that are visible include those that interact with the Earth's core – for example, PKKP, is the P-wave that travels through the mantle and outer core, reflecting once off the understand of the core-mantle boundary, before reaching the seismic station. Figure 9 shows the paths of various seismic phases that travelled from the site of the earthquake, denoted by the star at 0° to the station in Bristol, over a quarter of the way around the planet.

Here, we only show the phases that are most easily visible on the seismogram in Fig. 8 but it should be noted that there are other less-easily visible phases present in the seismic trace; there are a number of current research endeavours to study these more exotic signals. Teleseismic events such as this one provide our primary means of mapping the internal structure of the planet; they are how we know, for example, that the Earth has a core that is distinctly different from the mantle.

Seismic waves travel through the Earth like light travels through a prism; as the waves cross layers they encounter variations in material properties, which cause them to bend/refract, reflect or even change phase (e.g. P-waves convert to S-waves). Analysis of these changes in the seismic waves highlights the existence of the main layers within the Earth such as the solid inner core, the liquid outer core, the mantle and the comparatively thin crustal layer upon which life exists. The arrivals from



an event such as the Palu earthquake, which occurred at a relatively shallow depth of ~20 km, illuminate the existence of the
inner and outer cores when waves reflect off the inner core and either pass through or are diffracted off the outer core. Modern
day seismologists use arrivals from many earthquakes recorded by dense seismic networks to tomographically image the
Earth's three-dimensional internal structure, much the same way as medical imaging techniques are used to investigate the
internal structure of the human body.

Closer to home, the magnitude $M_w$ 6.8 that occurred approximately 44 km SW of Mouzaki, Greece on 25 October 2018, had
far fewer devastating consequences. Like the Palu event, this earthquake also registered on the tower seismometer and its
signal is clearly visible above the regular activity near the tower (see Fig. 10). Although smaller and less destructive than the
Palu event, the Greece event occurred closer to Bristol and was therefore more clearly visible. Data from this earthquake also
provides a treasure trove of information on the internal structure of the Earth. The P-wave and S-wave arrivals are clearly
visible phases that travel through the Earth's interior and are therefore referred to as body waves. The larger energy arriving
after 1000 seconds is associated with surface waves that propagate along the surface of the Earth. Surface waves are always
slower than body waves and are generally lower in dominant frequency.

## 2.4 Variations in signal amplitude due to local site effects

The amplitude of the seismic signal is a direct measure of the strength of vibrations affecting the seismometer. Signals that are
produced by the horizontal motions of the ground or wind should produce a stronger signal on the horizontal channels (HHE
and HHN) than they would on the vertical channel (HHZ). Likewise, signals produced by the vertical movement of the ground
beneath the tower should produce a stronger signal on the vertical channel than they do on the horizontals. A look at the data
from the tower shows that many of the signals recorded have higher amplitude on the vertical channel than they do on the
horizontals. This is somewhat surprising since the expectation prior to the launch of this experiment was that the natural sway
of the building, coupled with the impact of strong winds, would have greater effect on the horizontal channels than it would
the vertical.

At various points during the period that the seismometer was installed in the tower, there were other seismometers recording
in the Wills Memorial Building. Comparison of the data from these seismometers to that recorded on the tower seismometer
showed that the signal in the tower was being amplified on all channels. This is not surprising as shaking is expected to be
amplified in tall buildings particularly, when they are impacted by low frequency shaking that match their resonant frequency.
A building's resonant frequency is approximately inversely proportional to its height (Pratt et al., 2017). An example of this
signal amplification is shown in Fig. 11; here, the traces are filtered between 2 and 10 Hz to separate the earthquake signal
from the background noise. This figure shows a comparison between the signals recorded from the Lincolnshire earthquake
discussed above as recorded by a seismic instrument on the first floor (one floor above the street level) and GT01 in the WMB
tower. The amplitude of the data recorded in the tower is more than double that recorded on the lower level.





## 2.5 How does it compare?

While the day to day "seismic" activity around the WMB seems quite vigorous, it goes largely unnoticed by the city's inhabitants. This raises questions of how Bristol's daily "seismic" activity compares to seismic signals from real earthquakes.

Data recorded in the tower of both background noise and the bell peal were compared to seismic data from earthquakes of varying magnitudes recorded on stations in Hawaii. The Hawaiian earthquakes were recorded on seismic stations that were within 1 to 5 km from the earthquake epicentre. As illustrated in Fig. 12, the data from the tower are barely visible against the higher amplitude signals produced by real earthquakes. The smallest real seismic event on these plots is a magnitude 1 earthquake recorded at a station 1 km from the event. From the zoomed in view in the inset of Fig. 12, the amplitude of the

Bristol data (magenta and green) can be seen to fit neatly within the amplitude of the waveform for the magnitude 1 event (blue waveform). The Bristol data sit well below the vibrational levels of both the magnitude 3 (yellow) and the magnitude 6 (red) events. Placed in this context, it is clear why the activity in the city may go largely unnoticed. The amount of shaking produced by a magnitude 1 earthquake is seldom within the detection threshold of humans. However, as the event magnitude increases, it becomes easier for humans to detect a ground motion event in noisy settings even when the changes in magnitude

are relatively small. This is because the commonly used earthquake magnitude scale is a measure of the amplitude of ground displacement and is logarithmic; this means that an increase of 1 in magnitude equates to a tenfold increase in the amplitude of ground displacement. This relationship between earthquake amplitude and magnitude also carries over to the relationship between the energy released by an earthquake and its calculated magnitude. For instance, a magnitude 3 earthquake releases roughly 32 times more seismic energy than does a magnitude 2 earthquake.

## 2.6 Church bells ring during earthquakes

Historical reports of earthquake activity frequently include notes on church bell peals accompanying ground shaking during an earthquake, even in situations where the bell is not proximal to the earthquake epicentre. Such incidents were reported in Charleston, SC following the 1811 New Madrid earthquake which occurred on a fault approximately 977 km away from the church (LaCapra, 2011). Likewise, the spontaneous ringing of church bells was heard as far away as Paris following the 1755

Lisbon, Portugal earthquake (Penna and Rivers, 2013). Both earthquakes in the examples above were large but it is also worth analysing the effect of the energy released on structures far away from the epicentres. The Modified Mercalli earthquake Intensity scale estimates that bells ring with earthquake intensities of V to VI, which roughly corresponds to a magnitude 5 on the Richter Scale. The relation between earthquake magnitude and energy can be written as



$$\log_{10} E = 5.24 + 1.44M, \tag{1}$$

where E is the energy released by the earthquake and M is the moment magnitude of the earthquake magnitude (USGS Science Center, n.d.). Therefore, a magnitude 5 event corresponds to an energy release of $2.75 * 10^{12}$ J.

This calculation shows that there is clearly a large amount of energy released by a magnitude 5 earthquake, but this number only relates to the energy released in the immediate environment of the fault that generated the earthquake. As the seismic

energy propagates out from the earthquake epicentre, it dissipates; this is referred to as geometric spreading. The formula for estimating the relationship between the energy dissipation and earthquake distance is given in Eqs. 2 and 3 for surface waves and body waves (P and S phase) respectively

$$E \propto \frac{E_0}{2\pi r} \tag{2}$$

$$E \propto \frac{E_0}{2\pi r^2} \tag{3}$$

where, $r$ is the distance from the earthquake epicentre measured in metres and $E_0$ is the energy at the earthquake source. The surface waves occur later in the wavetrain than the body waves and tend to be of higher amplitude (see, for example, the Greek earthquake and the signals arriving after 1000 s in Fig. 10). These waves tend to cause the most shaking and therefore, the most damage.


The 1811 New Madrid earthquake was estimated as ranging from magnitude 7.5-7.9. As an example, we can estimate the amount of seismic energy that arrived in Charleston, SC. The surface wave energy reaching the city was about 6 orders of magnitude larger than the body wave energy but was only equivalent to the energy released by a proximal magnitude 3 earthquake. Magnitude 3 earthquakes are considered weak events that cause little to no structural damage. Therefore, for

ringing to occur under these conditions, other factors may have been influential. For instance, church bells are often left in the "ready" position (Woodhouse et al., 2012), which reduces the inertial forces that must be overcome to swing the bell. It would likely be easier to move the external clapper during an earthquake than it would be to move the bell and its internal clapper. A third factor which may have influenced the ringing of the church bells during an earthquake is the amplification effect of the bell tower. Research by (Blakeborough, 2001) showed that while the ringing of the bell is dependent on the mechanics of the

bell and the tower in which it is hung, the peak ground acceleration (PGA) typically required for unplanned ringing needs to exceed the PGA of a magnitude 5 earthquake. From our own experiment, illustrated in Fig. 11, there was at least an order of magnitude increase in amplification between the signals measured on the tower compared to the seismometer on the lower floors of the building. The signal measured in the tower was roughly doubled that measured on the lower floor but would not be enough to amplify the shaking from a magnitude 3 to a magnitude 5 event. It therefore, remains unclear the exact mechanism

responsible for producing the historical unscheduled ringing of church bells reported following large earthquakes.



## 2.7 Transforming seismic data into art

Monitoring ground shaking and earthquake activity is mostly the job of scientists and researchers, whose areas of expertise connect with the distant global environment. And for those who live near an area that directly experiences the Earth's tectonic
activities, the contact is only too real and impacts daily lives. But for many where this impact is not usually felt or experienced, being made aware of the constant noise and movement occurring under our feet can be surprising. The creative aspect of this project thus experimented with performance in order to encourage public engagement with the energy, force and distant origins of particular sources of sound; onlookers and audiences could fast track the "scientific explanations" of the unsettled Earth's activity and instead perceive it in a direct and visceral way. After hearing recordings captured during the ~1-year deployment
of the seismic station in the WMB, for example, the choreographer Will Pegna was inspired to create a durational dance piece, ***All Terrain Training***, performed in Peckham in an artist-led space. A photo captured during one of the acts of this performance is shown in Figure 13. The particular source of the recordings in this case came from a $M_w$ 7.4 earthquake on 28 September 2018 near Sulawesi Island, Indonesia which is ~12,200 km from Bristol and a $M_w$ 7.9 on 6 September 2018, near Viti Levu Island, Fiji, ~16,000 km away. These recordings were amplified and mixed live for the event by Shirley Pegna. To convey the
power of the low audible and the low inaudible vibrating frequencies, the cone of a 22-inch subwoofer was employed along with speakers to create a surround sound and immersive effect for the audience members (see subwoofer cone vibrating: https://vimeo.com/389225640 accessed: 08/04 2020). A second performance piece was also made, inspired by the earthquake sounds mentioned above. This piece, ***Earth Din***, presents these sounds via three sub woofers and four speakers placed in a large space where an audience can walk around amongst the speakers and be immersed in the sound and vibration. To create
further interest by embellishing the experience of the natural sound, the piece also requires two musicians intermittently playing cello and double bass. The two musicians work with the sounds reacting to them and each other in the moment, improvising the sounds they add to the soundscape. The sharing of the sound and vibration from our planet to an audience via both these expressive performances thus creates another way to experience the Earth activity data recorded from the Wills Memorial Building tower.


In particular, the set of vocabularies we have for size, scale, distance and time need to be re-imagined, processed and re-assessed when thinking about the vastness of the Earth for those of us not used to thinking in this way. With this in mind, the third artwork coming from the ideas raised by this interdisciplinary project dealt with time, and explored the notion of geological time. The birth of planet Earth, its existence, and its projected demise spans a very different duration compared to
our experience of time day to day. Shirley Pegna's ***The Rock Record***, an object for a conceptual artwork, plays on the term used in geology to refer to the stratigraphic record that denotes how geologic time is captured in a rock sequence, and is in this case a rock into which is etched sound wave data from the recordings captured from the WMB. Like the copper plate into which sound waves of music are etched in the process to press a vinyl record, the etching we are currently creating will sit embedded in a record (LP) made of rock. During this process, the seismic data recorded by the seismometer is first converted





to a sound file which is then be etched into a record, not made of vinyl but of very durable rock. This token design copy of a contemporary media player will outlast the 21st century objects made of degradable vinyl, wood and plastic and possibly outlast even the span of human existence in rock form. In theory, it may even last into geological time itself.

## 3 What determines how people respond to ground shaking?

As discussed above, the Earth is constantly moving and vibrating around us though this motion is a frequently accepted part
of our daily lives and often ignored. This general acceptance of ground motion leads to the question of what determines human responses to Earth vibrations. The eventual acceptance of the theory of plate tectonics, some fifty years ago, ushered in a new scientific paradigm that explains the driving forces behind earthquakes (McKenzie, 1977; Oreskes, 1999). The Earth through its frantic effort to cool itself through convection and conduction, has created and destroyed entire tectonic plates many times through its history. However, at various points throughout our human history, earthquakes have played a central role in human
thought and development. From early ideas concerning punishment or exercise of divine wrath, to the facilitation of dynastic changes to enable modernisation, earthquakes have been explained through folklore, religion, and philosophy (Winchilsea, 1669; Mallet, 1728; Ponton, 1872). The Great Lisbon Earthquake of 1755 is a famous example of an earthquake accelerating divisions between religion and science, leading thinkers and poets to question the benevolence of a divine god. Indeed, as Juengel (2009) summarises, "Lisbon has of course long loomed as the world historical event that cracked the foundation of
Enlightenment Optimism, driving Voltaire to abandon Leibniz, Pope, and the rightness of what is and to embrace Bayle's more profound doubt" (see also Nicols (2014)).

For historical earthquakes from the "pre-instrumental era" prior to the last decade of the nineteenth century, the only method of study is to use intensity, which is a number quantifying the degree of shaking at a particular point. In the case of the
earthquake of 6 April 1580, for example, which was felt throughout a large portion of England, northern France and the Low Countries, and is now thought to have had an epicentre in the Straits of Dover, a rich multi-lingual textual record of sources, including letters, new pamphlets, prayers, treatises and even satires, bears witness to the effects of the shock (Neilson et al., 1984). These sources include not only vivid literary responses to the earthquake, all the more fascinating for the different generic and ideological frameworks through which they are presented, but also document the particular effects felt in specific
locations, from falling masonry, to bells ringing, and theatres shaking (Totaro, 2017). In his discussion of the event, the physician and translator Twyne (1580) wrote of how 'the very shaking caused the bells in some steeples to knoll a stroke or twain', and that 'the tops of half a dozen chimneys in London were cast down'. As someone who did not feel the ground moving personally, he was keen to address how not everyone felt the effects of the earthquake equally:

Some that were above in their chambers, commonly judged that some violence had been done to their houses beneath.
450       Some that remained below, found fault with tumbling and trampling above. Some imputed the rattling of wainscots to rats and weasels: the shaking of the beds, tables, and stools, to dogs: the quaking of their walls to their neighbours





rushing on the other side. And as their opinions were sundry, so were their speeches thereupon diverse, until a common conference being had, they were resolved upon their common case and danger. For many not trusting to their own judgement, and partly also moved with fear, ran out into the streets to know if the like had happened unto

others.

Twyne's contribution to the literary record of the 1580 earthquake is noteworthy owing to the way it captures the reactions of ordinary people and how they speculated about the causes of the ground shaking, looking first to everyday explanations for felt motion, and then, having realised the novelty of the occurrence, to a kind of community consensus in order to understand more fully the scale and nature of what has happened. As Neilson et al. (1984) have shown, where multiple testimonies of this

kind exist, details concerning felt effects at particular locations can be used to create intensity maps, and to estimate the focal depth and magnitude of events, in this particular case resulting in an estimated Richter magnitude in the range $M_L$ 6.2-6.9.

From the perspectives of the humanities, then, responses to the ground moving can be seen to be determined by how lived experiences are processed and understood within pre-existing cultural, theological, linguistic, philosophical and narrative

frames. In historical reports it seems to matter whether the right language can be found to articulate the experience of the ground shaking, and this can often involve analogy in order to communicate the precise nature of the sensations experienced, which can be likened to a cognate encounter of unsteadiness or vibration. Perhaps more importantly, the use of literary genre to construct a deliberate interpretative frame asks us to consider how genre gives shape to different kinds of temporal relationships and possible endings, owing to how genres can be used to determine narrative expectations by connecting

causality, action and consequences in particular ways. In Europe, representations of earthquakes have been historically framed not only by tragic modes which speak of hubris, divine retribution, and even apocalypse, but also through providential models, including those of recuperation, where the earth quaking takes on the significance of a divine admonition, calling the faithful both to repent and to give thanks for their deliverance. These literary and theological frameworks co-exist in generative ways with local folklore, eye witness testimony, and the rationalisations of science, which in Europe, from the late sixteenth-century

onwards, grew out of the narrative methods of natural philosophy and drew on the authority of classical treatments of earthquakes and their causes by writers such as Aristotle and Lucretius (Prewitt, 1999; Passannante, 2008). In early modern Europe, c.1500-1800, surviving textual responses to natural disasters also reveal active transnational networks of printed news – an early version of our contemporary rapid mediation of such phenomena – and the ways in which environmental events, including floods, volcanoes, and earthquakes, could be used for political ends, prompting the state-authorised publication of

guides for public worship (Mears, 2012; Caracciolo, 2016). The historical records of natural disasters are also useful for dating literary works. The composition of William Shakespeare's *Romeo and Juliet*, for example has traditionally been calculated on the basis of an earthquake reference made by the character of Juliet's Nurse (Thomas, 1949; Dodson, 1950).

To write in response to catastrophe, then, is to engage imaginatively with the limits of human reasoning and to formulate

representational, and even aesthetic, modes capable of handling epistemological change (Heringman, 2003). As Totaro (2017)



observes of the 1580 event, "the earthquake imprinted the body-mind and altered all other sensory-related terrain immediately": reactions which transformed the event into "a subject of epic proportion". To connect the scale of the earth quaking to the epic mode is perhaps to consider what it takes for a culture to live and rebuild in the aftermath; yet, for writers of the eighteenth-century responding to the Lisbon earthquake of 1 November 1755, the magnitude of the destruction was connected to the

sublime. As William F. Fleming (2005) writes, summarizing the hitherto unparalleled scale of the event, "within the space of five or six minutes, at least thirty thousand people died; the city was reduced to ruins, and the earthquake itself was accompanied by fire and the flooding of the Tagus River". The poem written by the French Enlightenment writer Voltaire (see Voltaire and Fleming (2005)) in response to the calamitous events responds to the mutable, tremulous nature of the Earth, and includes the author's struggle with the long-held but now newly troubled idea that we live in the best of all possible worlds:

495            I can't conceive that 'what is, ought to be,'

           In this each doctor knows as much as me.

           We're told by Plato, that man, in times of yore,

           Wings gorgeous to his glorious body wore,

           That all attacks he could unhurt sustain,

500            By death ne'er conquered, ne'er approached by pain.

           Alas, how changed from such a brilliant state!

           He crawls 'twixt heaven and earth, that yields to fate.

           Look round this sublunary world, you'll find

           That nature to destruction is consigned.

For Jörg Trempler (2013),  the Lisbon earthquake not only ushered in a "new condition", owing to how the long-held "philosophical tendency that regarded the world as the best of all possible worlds" suddenly "found itself incapable of explaining away the destruction of a capital city", but also prompted the invention of "catastrophe" as an iconographical tradition. In the transient world figured by Voltaire it is the fate of contemporary humankind to walk or merely "crawl" upon the earth, rather than to be raised above it, and to be subject to the shaking, impermanent motions of a "sublunary" existence.

The impact of the earthquake, as addressed in such terms, was to reveal the sublime power of nature's latent destructive capacity. Indeed, to narrate "catastrophe", as Juengel (2009)observes, is thus to confront the "telltale shuttling between measure and melancholy" that "structures the epistemological work of the postdisaster". As he explains, "the word catastrophe already possesses its own strophic or figural quality, for etymologically the term's Greek origins signal an "overturning" of the order of things; thus "catastrophe" becomes what we name a suffering heretofore unthinkable".


A similar scaling of response, used to confront the "unthinkable", is seen when comparing large earthquakes between regions in modern society. In general, people living in earthquake prone areas in developed countries like Japan tend to be more comfortable with large earthquakes than those living in developing but earthquake prone countries like Haiti. This relative comfort is imparted by confidence in the engineering and disaster management in place in developed countries (Kahn, 2005).



However, this confidence can potentially be a double-edged sword leading to complacency which may result in significant loss of life and resources as illustrated in the aftermath of the 2011 Tohoku earthquake ($M_w$ 9.0 – 9.1). The ensuing tsunami led to the destruction of the Fukushima Daiichi nuclear power plant and a subsequent moratorium on nuclear energy in Japan amongst other regulatory changes. In the less developed country of Haiti, the 2010 $M_w$ 7 earthquake affected over 2 million people, damaged 80-90% of homes near the epicentre and had a death toll of well over 200,000 people (DesRoches et al.,
2011; Rebekah Green and Scott Miles, 2011). The impact of the earthquake in Haiti was likely due to two main causes; unlike Japan, Haiti's economy precludes the necessary level of engineering and disaster preparedness to minimize earthquake damage. Additionally, while a similar magnitude earthquake had struck the country in 1945, the memory of that prior event had most likely faded from the collective memory leading to an underestimation of the disaster potential. We are left, then, with the idea, as Fountain and McLaughlin (2016) observe, that "there are no purely 'natural' disasters. Once a hazard phenomenon –
flood, drought, tsunami, earthquake, typhoon/ hurricane, geothermal activity, or other catastrophe – intersects with human society, any purported naturalness dissolves. All disasters are enmeshed in human worlds involving economic, cultural, political and religious concerns".

In recent years, humans have also become acutely aware that their activities and interactions with the sub-surface of the Earth
induce earthquakes, sometimes in magnitude at dangerous levels. However, human acceptance of such *induced seismicity* is variable, depending on the contexts within which such activities occur, which are shaped by familiarity and economic goals. On one hand, the United Kingdom has experienced coal-mining induced earthquakes since the Industrial Revolution and the start of large-scale underground mining (Redmayne, 1988; Wilson et al., 2015). On the other hand, much smaller induced earthquakes have paralyzed the petroleum industry's efforts to produce domestic gas from shale formations in places such as
Preese Hall near Blackpool in Lancashire. Ground motion associated with a stadium concert or sports events can be as large as that associated with sub-surface activity (e.g., fracking), but the latter causes far greater alarm. Indeed, induced seismicity associated with fracking has proven to be very controversial, especially in the United Kingdom. During the fracking process, large volumes of high-pressure water are injected into deep rock formations that contain gas trapped in and impermeable matrix. The water stimulates a fracture network through the generation of tiny earthquakes – the energy produced by these
small earthquakes is roughly equivalent to dropping a book (British Geological Survey, 2008). However, in rare cases, the water interacts with large pre-existing critically stressed faults. This water-fault interaction in turn leads to the generation of larger and felt seismicity. Such was the case in April 2011 when a $M_w$ 2.3 was triggered in Preese Hall which led to an 18-month suspension of shale gas production in the United Kingdom (Clarke et al., 2014). It should be noted that this induced earthquake occurred at a depth of 3.6 km and would have only produced surficial ground motion of slightly high amplitude
than a train crossing a viaduct (see Horleston et al. (2013)). When operations resumed in a nearby site (Preston New Road) in 2019, exploration was subject to a very strict traffic light system. Within this traffic light system, following a magnitude 0.5 earthquake, the operator is required to halt any further injection, reduce pressure in the system and monitor for any new seismicity before operations may potentially resume (Oil and Gas Authority, 2013). On the 26 August 2019, hydraulic fracture

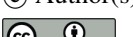



stimulation near the Preston New Road site in Lancashire triggered a ML 2.9 earthquake, again leading to a moratorium on
fracking in the UK (Kettlety et al., 2020).

In contrast, fracking has led to much larger earthquakes in Canada (up to M 4.1 in Fox Creek, Alberta (Schultz et al., 2017))
but there is a lot less restrictive traffic light system in place in Canada and perhaps lower public concern. The difference in
reaction may be threefold: a) Albertans are much more accustomed to the petroleum industry, as for many it is in fact their
livelihood; b) the region where shale gas activity is taking place in Canada is less populated than Lancashire; and c) there is
natural seismicity of similar magnitude in this part of Alberta, providing residents with a measure of familiarity with this level
of ground shaking. The occurrence of large induced earthquakes is not solely associated with fracking. There are numerous
anthropogenic causes of seismicity including mining, oil and gas activity, water impoundment via dams and weirs and
geothermal energy production (Foulger et al., 2018). While these activities normally lead to minor amount of micro-seismicity
(earthquakes less than Mw 2.0), there are notable exceptions. For example, in 1967, the construction of the Koyna hydroelectric
dam in India led to Mw 6.6 earthquake (Berg et al., 1969; Gupta et al., 1969). Oil and gas activity in Uzbekistan led to a Mw
7 event in 1976 (Foulger et al., 2018). Recently, the American state of Oklahoma has seen a dramatic rise in seismicity which
is attributed to sub-surface wastewater injection associated with oil and gas extraction. Here, the largest events were greater
than magnitude 5 and led to damage to several homes and businesses as well as a few injuries (Keranen et al., 2013).

**4 Closing Thoughts**

During the final stages of writing this paper, the coronavirus reached the UK and we all went into a period of extended
lockdown and isolation. As everyone stayed at home, except those providing essential services, the anthropogenic noise in the
country dropped to levels not experienced in decades. In additional to traditional journal sources (e.g. Gibney (2020)),
seismologists across the world have been sharing on social media visual graphics that illustrate reductions in seismic noise
across  most seismic networks. Figure 14 is on such example, produced by Dr. Stephen Hicks at Imperial College in London.
It shows the seismic noise on two UK networks (BGS and Raspberry Shake), comparing levels before and after lockdown. It
also shows correlation with UK retail and recreational activity.

The British Geological Survey (BGS) operates a dense network of seismic stations, using instruments similar to those we
deployed in the Wills Memorial Building. These stations are deliberately deployed in quiet settings. Nevertheless, a clear drop
in background noise is visible starting in late March 2020 (Fig. 14). In recent years, seismometers have been deployed that
cost a fraction of the price of conventional seismometers, but with only a slight degradation in instrument sensitivity. The
Raspberry Shake instrument (raspberryshake.org) is a very popular example. Unlike the more conventional seismometers,
these instruments use a MEMS (micro-electo-mechanical0systems) sensor which fits on a small chip. Their use is a good
example of citizen science in that the data are shared publicly so that scientists across the world can use the data. They are





deployed in people's living rooms, sheds, etc. and there are currently more that 50 Raspberry Shake seismometers in operation in the UK (raspberryshake.net, 2020). Figure 14 shows how the seismic noise dropped on both networks, but especially the Raspberry Shake network, during lockdown. There are also clear variations in noise between weekdays and weekends.

The ground beneath our feet may be unusually quiet, then, at this precise moment of writing; however, a glance at the language of recent headlines reveals that "seismic shifts" are occurring in other terms, and in forms of activity across all areas of daily life, from consumer habits and corporate culture to educational practices and trends of global politics. Indeed, our cross-disciplinary conversations prompted an increasing awareness of how we use different vocabularies, with different resonances, to articulate the phenomena of earthquakes and how these vocabularies resonate within everyday speech. From a literary

perspective, it is striking how some of the most commonly used vocabularies carry metaphorical currency, having either been absorbed into colloquial ways of speaking of non-specific adverse events, or, more complexly, tasked with operating in intrinsically figurative ways themselves, retaining the memory of an emotional, moral, biological or physical charge: aftershock; stress-trigger; rift; fault; tremor; swarm; cascade; shadow zone. The increasingly over-used idiomatic term "seismic shift", for example, used to connote conceptual change across all aspects of human experience, has been increasingly

commonplace in our public discourse since the 1980s, as the Google Ngram Viewer (Lin et al., 2012) bears witness (see graph linked here: https://tinyurl.com/ydaob65o). As Taylor and Dewsbury (2018) write, synthesising the arguments of Lakoff and Johnson (2008)'s *Metaphors We Live By* in their discussion of metaphor in science communication, "metaphors are not mere linguistic embellishments. Rather, they are foundations for thought processes and conceptual understanding that function to map meaning from one knowledge and/or perceptual domain to another". During this period of lockdown, we deploy the

language of seismic activity – its shifts and its impacts – to give shape to something that we do not yet fully comprehend.

For the project's duration, we also found ourselves drawn to a set of vocabularies related to the sonic potential of vibrations, whereby the earth can be thought to sound like a giant bell: harmonics; tones; fundamental modes; whispering gallery waves. Indeed, the bell like sonority of the Earth, our "blue planet", resembles a resonant chamber where seismic waves travel through

and around its interior. The comparison is made with the Whispering Gallery in the roof of Saint Paul's Cathedral, where sound waves are reflected horizontally round the walls circle, and bounce around multiple times because the angles involved are so slight (Fitzgerald, 2016). Interestingly, seismic phases can propagate around the inside of the planet in a similar fashion, clinging to the underside of the Earth's core-mantle boundary, some 3000 km beneath our feet. Our data from the Wills Memorial Building tower at Bristol University showed evidence of seismic waves coming not only from the tower bell, Great

George, and local city impact, but also extremely distant impacts travelling to the instrument from seismic activities on different continents. Our response to the data realized in sound and vibrations from the WMB tower seismometer could be described as hearing "music as audible physics" (Young, 1998), where the waves travel around the inside of the Earth are described as resounding and reverberating. In a reciprocal way, such realizations can also be used to help scientists interpret





seismic data (e.g., Paté et al. (2017)). Transforming earthquake signals into audible sound help us to better understand the scale
of seismic shifts in the Earth (see, for example, http://www.seismicsoundlab.org).

The technology, namely the seismometers and the geophone used either in, or within the vicinity of, the WMB tower to "listen in" to the ground, acted like bionic extensions to our human ears. Seismic waves reached the instruments from great distances by travelling through the ground, and so being able to hear and experience the waves altered our perception of what goes on
under our feet and in our surroundings, by expanding the reach of our hearing and extending our own auditory map further than the limitation of our anatomy: our planet might be unsettled but we are implicated in its motion in complex ways, our own bodies forming part of a network of sensing instruments. The ear has been described as "half anatomy and half imagination", by writer and cultural critic Connor (2010), who describes how our perception of sound enhances our understanding when we can only imagine a sound's location but cannot see its source. Our technology also picked up different
frequency waves, some audible and others inaudible to our ears, prompting a thought experiment to imagine what we could not hear with our human ears when perceiving seismic waves pitched below our auditory range. Yet, if we could not hear the waves as audible sound, they could still provide an impression of our surroundings. Musician Evelyn Glennie, who has a hearing impairment, comments in her *Hearing Essay* that "hearing is basically a specialized form of touch" (Glennie, 2015). When we could not hear but could only feel the vibration of waves, amplified through a speaker, we were still receiving an
impression of the energy coming from the earth.

Connor (2010) describes sound and place as being intrinsically linked, where a "sound is the space in which it occurs" and "sonic essence inheres in spatial accident". In our discussions, we became fascinated by extending the notion that people "site" themselves in place by connecting to sound, to the idea that attending to vibrations in the earth requires people not only to
"site" themselves in relation to their local environment, but also to their global environment. John Luther Adams, for example, in his work *The Sonic Geography of Alaska*, collates different elemental signals from the natural environment (Young, 1998). His reach is also global and he too states that "sound is a way of touching at a distance". From our own place of relative safety, then, we re-evaluated our understanding of the ground beneath our feet and attended to what the vibrations travelling though it can tell us about the domains and limits of our situated disciplines and institutional habits. Our experiment to discern the
motions of our unsettled planet, which focused on the bell Great George, thus ran alongside a second experiment, which tested novel collaborative structures of research and aesthetic practice. The result has been an unsettling of accustomed methods, stress-testing the disciplinary foundations on which we build.

Data Availability: The raw data utilized in this study will be archived with the National Geoscience Data Centre (NGDC).

Author Contributions: All co-authors contributed to the study and in writing the manuscript.



Competing Interests: Authors declare they have no conflict of interest.

**Acknowledgments:**

This work was funded and facilitated by the University of Bristol's Brigstow interview as part of their "Living well with uncertainty" initiative in 2018. O. George and J. M. Kendall were funded by NERC Grant NE/R018006/1. We gratefully
acknowledge support from the sponsors of the Bristol University Microseismicity Projects (BUMPS). We thank Dr. Stephen Hicks for generating the plot in Fig. 14.



Figure 1: Great George Bell (b) sits within the belfry of the Wills Memorial Building pictured in (a) on a typical weeknight. The broadband seismometer (c) installation setup in one of the alcoves above the bell. The instrument itself lies beneath the black dome with the Brunel rubber duck (geophysics group mascot) on top.




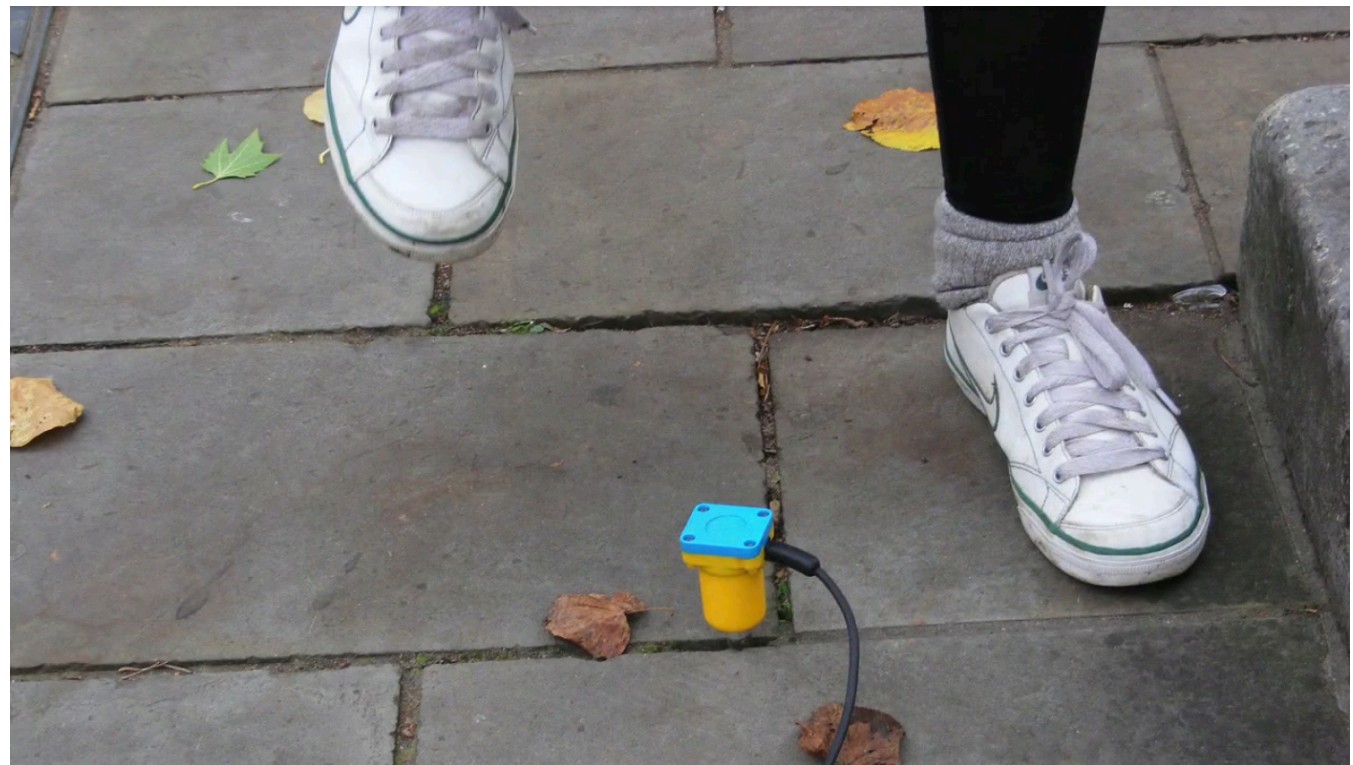

**Figure 2: Workshop participant jumping near geophone set up to record foot traffic near the Wills Memorial Building**




**Figure 3: 24 hrs of continuous seismic data recorded on three channels (HHE, HHN, HHZ) at the seismic station GT01, which is housed in the bell tower of the Wills Memorial Building, The orange box on HHE (top) trace highlights the bell chimes at 10 PM (9 PM UTC). The red box shows the much larger bell ringing for the Charter Day commemoration on 23 May 2018. Both highlighted signals are explored further in Fig. 4. The numbers in the upper right corner of each trace indicates the absolute maximum amplitude**
**of that trace in counts.**





**Figure 4: Zoomed in spectrogram plots of the signals highlighted in Fig. 3. Plot a) shows the 10 PM (9 PM UTC) chiming of Great George in which each peal is clearly visible particularly in the 15-25 Hz range. Plot b) shows a spectrogram of the charter day chimes; note that the time scales are different and the colour range on plot b) is large than in plot a) to prevent colour saturation.**






**Figure 5: Helicorder recordings showing seismic signals (A) during a typical weekday near the WMB tower and (B) during a Saturday evening near the tower. During the week, while never fully quiet, activity tends to wane at night around 10 PM in contrast to a weekend night when activity is heightened until about 3 AM.**







**Figure 6: Spectrograms of the time-domain data shown in Fig. 5. The spectrogram is displayed in the bottom portion of plots: a) a weekday, and b) a day on the weekend. The top portion of each plot condenses the 24hrs of waveform data shown as a helicorder record in Fig. 5 into a single trace. The spetrograms show that there is strong energy in the 10-20 Hz frequency band throughout the day; this energy is almost continuous on the weekends (Plot b) but wanes slightly overnight on a weekday (Plot a).**









**Figure 7: Lincolnshire earthquake (a) as recorded by station LMK (~5 km from the source), and (b) the later arriving signal at station GT01 in the WMB tower (~300 km from the source). The HHZ traces show the vertical components of ground motion and the HHE and HHN traces show the horizontal components, which are oriented E-W and N-S, respectively. Note the difference**




between signal duration on bother seismometers as well as the difference in peak amplitude given by the numbers in blue in the top
       right corner of each trace. The estimated arrivals of the P and S waves are indicated on both plots. Note that the zero time in each
       case is arbitrary and the time scale is different for the two seismometers.

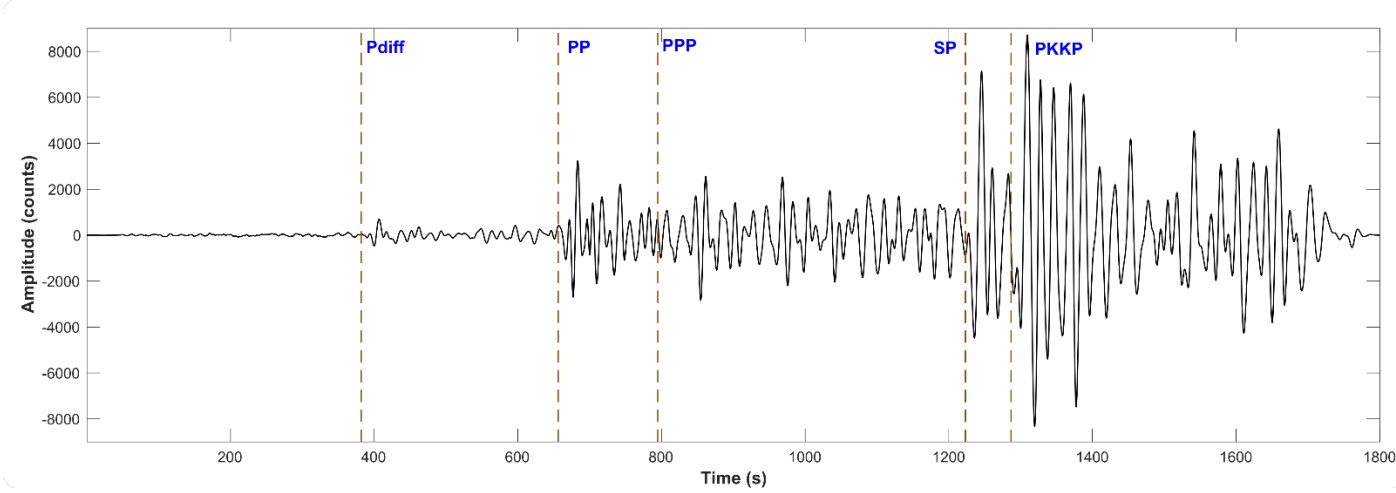

**Figure 8: Filtered seismic waveform showing some of the seismic phases visible from the 28 September 2018 Palu, Indonesia
       earthquake. The vertical-component waveform is bandpass filtered between 0.01 and 0.1 Hz to isolate the signals of interest. The
       earthquake occurred at a depth of 20 km and was ~12,200 km from the recording station (GT01) in the Wills Memorial Building
       (source USGS (2018)).**





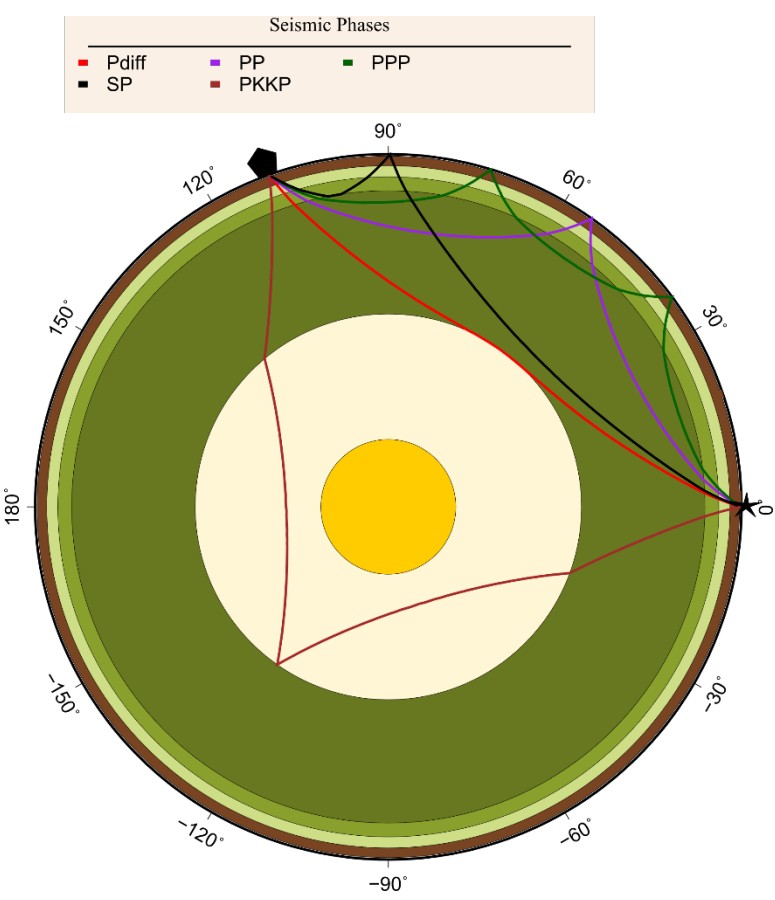

**Figure 9: Plot showing some of the ray paths that seismic waves travelled from the source in Indonesia to the seismic station set up**
**in the WMB tower. The star shows the earthquake epicenter while the pentagon denotes the station location. The inner core is shown in yellow, the outer core in beige, the mantle in green and brown in the image. The thin crust on which we live is the black outer layer, which is barely visible on the image. The arrivals of these phases are marked on the seismogram in Fig. 8.**





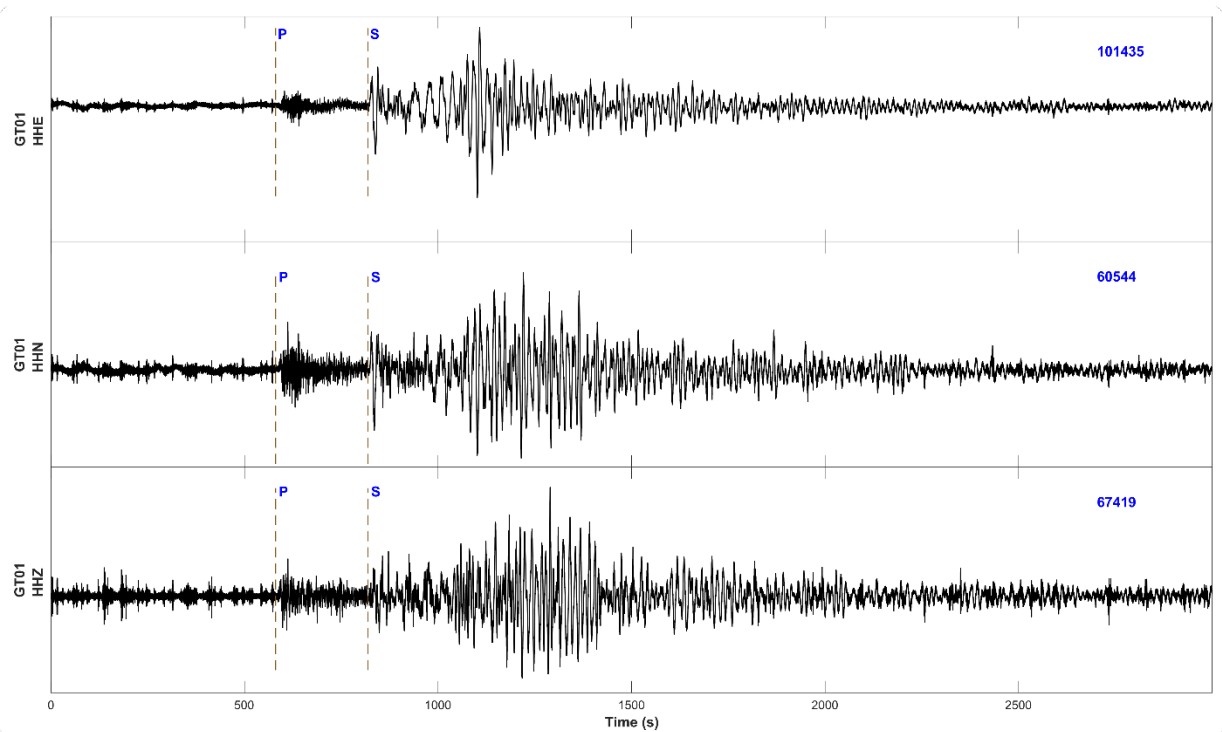

**Figure 10: Seismic recording of the magnitude 6.8 earthquake that occurred 44 km SW of Mouzaki, Greece on 25 October 2018. The earthquake occurred at a depth of 14 km and was ~2,400 km from the recording station (GT01) in the Wills Memorial Building. The HHZ trace shows the vertical component of ground motion and the HHE and HHN traces show the two horizontal components, which are oriented E-W and N-S, respectively. P- and S-wave arrivals are indicated; the stronger lower-frequency energy that arrives after 1000s shows the arrival of the surface waves.**





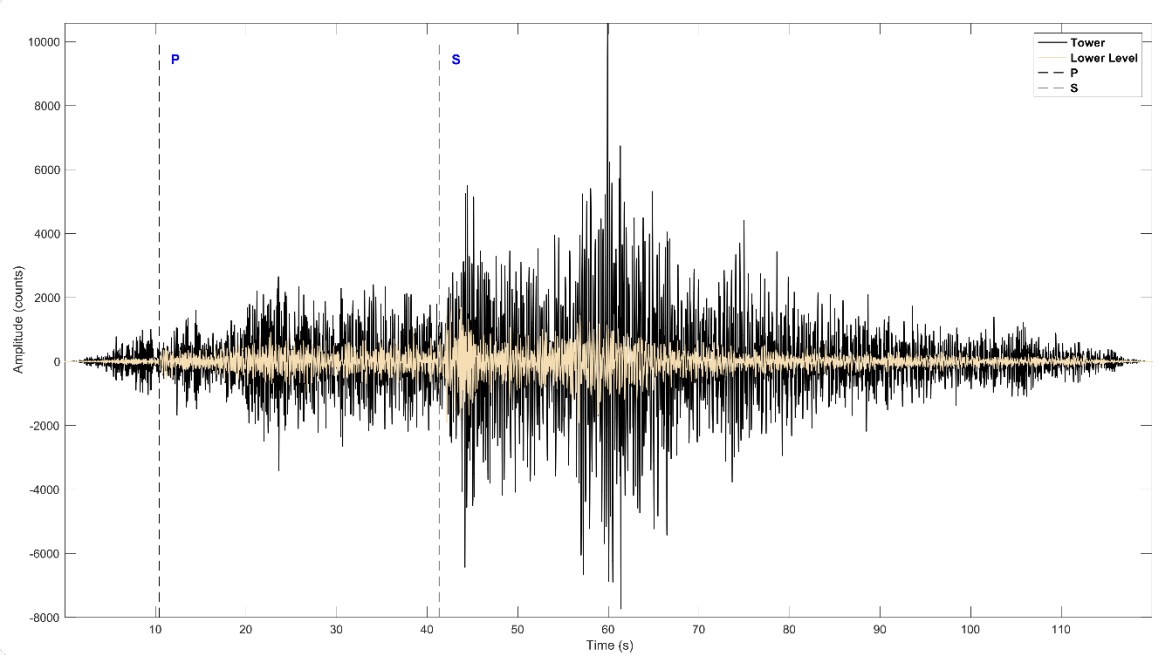

**Figure 11: Seismic signals from the Lincolnshire earthquake recorded on one of the seismometers on the first floor of the WMB (beige) overprinted on a waveform of the same event recorded on station GT01 (black) in the WMB tower. The signal from the tower is more than double the amplitude of that recorded on the first floor of the building. Both traces were bandpassed filtered between 2 and 10 Hz.**



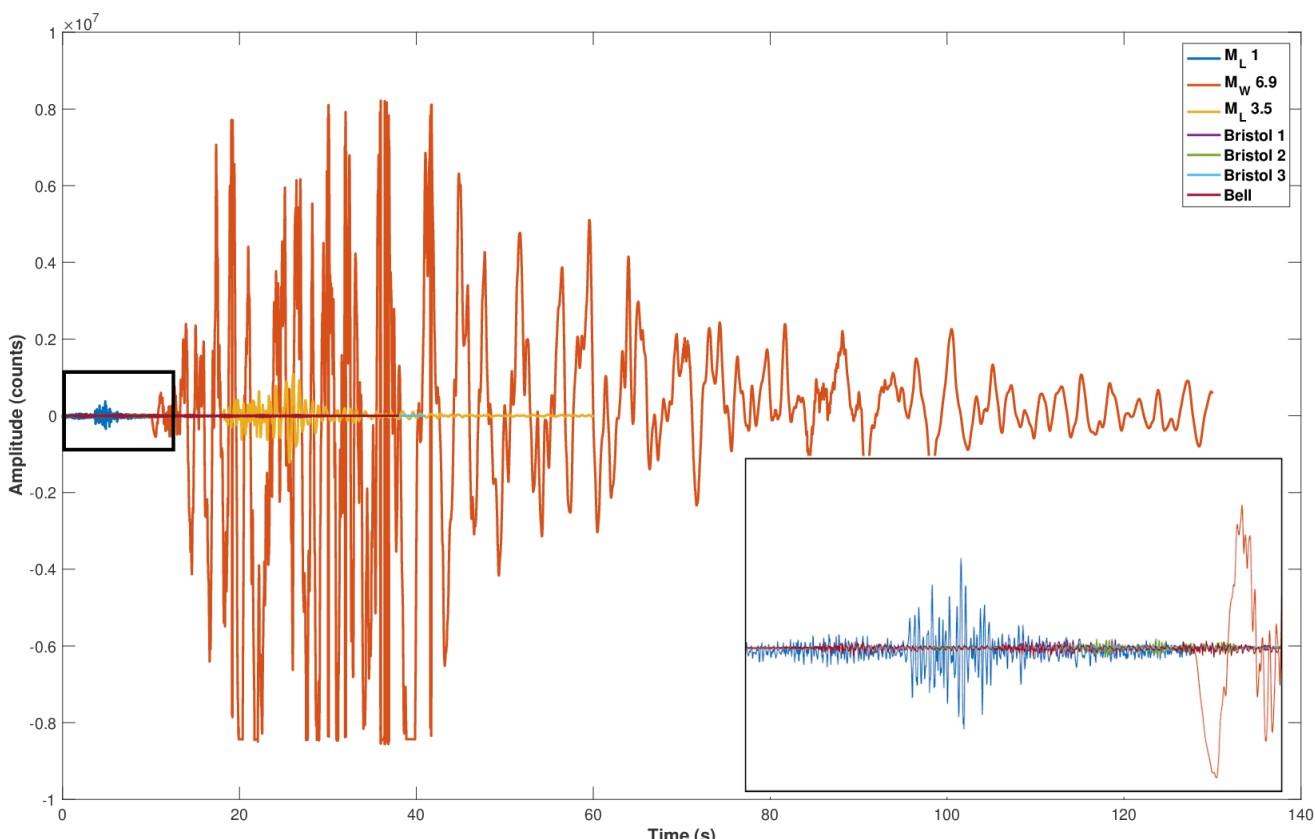

**Figure 12: Comparison plot of different seismic signals from various sources recorded on stations in Hawaii (signals labelled $M_L$ and $M_w$) and on GT01 in the WMB (signals labelled Bristol or Bell). Lower right inset shows a zoomed in view of the same signals highlighted by the black rectangle in the main figure. From the inset image it can be seen that the background noise and the bell ringing (green and magenta signals, respectively) recorded in the WMB tower fits neatly into the signal of a local magnitude 1 event (dark blue) recorded on a station 1 km from the earthquake source in Hawaii. Given that magnitude 1 events are seldom felt, the inset makes it clear why the background activity around the tower goes largely unnoticed.**




**Figure 13: Artists perform a durational dance piece entitled All Terrain Training choreographed to sonified seismic data captured in the Wills Memorial Building tower that was amplified and live mixed during the event. The choreographed movements depict the collision, shearing and spreading of the tectonic plates that produced the earthquakes to which the piece was performed. Photo credit: Rocio Chacon**






**Figure 14: a) Plot of average seismic noise recorded in the UK by the British Geological Survey (BGS) seismic network (blue) and the Raspberry Shake (RS) network (red) shows a marked decrease in seismic noise levels on both networks since the Covid-19 lockdown began on 23 March, 2020. This plot shows a decrease in average noise level of ~0.6 nanometres and ~15 nanometres on the BGS and RS network respectively between the peak noise levels in early February and the government lockdown in late March. b) The Google mobility data also depict a similar decrease in activity in the UK associated with retail and recreation activity trends post-lockdown. This plot shows that UK recreation and retail activity dropped by ~80 percent post-lockdown. Figure created by Dr. Stephen Hicks of Imperial College, London.**





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
