# Peer review of "Good Vibrations: Living with the Motions of our Unsettled Planet"

_Geoscience Communication, 2020_

## Referee Comment (RC1) · Paul Denton (Referee) · 8 Jul 2020

General Comments This is a fascinating study which brings together seismologists, artists and researchers from the schools of English, Earth Science and Art History at the University of Bristol. This unique variety of expertise allows for the analysis of seismic signals and their perception by people in ways that have not been considered before, bringing a refreshing perspective to seismic studies from a new angle.

Specific comments In section 2.5 in the last couple of sentences there is some confusion about the relationship between earthquake magnitude, signal amplitude and event energy release. As written it implies that the relationship between magnitude and amplitude (x10 for one unit) is similar to that for energy but then goes on to say that this

is a x32 per unit magnitude relationship. This could be clarified to differentiate better between the two relationships.

In section 2.6 (line 351-352) it states that a Modified Mercalli intensity value of V -VI corresponds to a magnitude 5 on the richter scale. In order to avoid confusion between intensity and magnitude concepts it would be better to modify this statement along the lines of " ... corresponds to the shaking experienced close to the epicentre of an earthquake with a magnitude of 5 ... "

In section 2.6 (line384) it states that the mechanism for bells ringing remains unclear. I would have liked to see a discussion at this point about the possible effects of resonance between the seismic wave frequency and the natural oscillation frequency of either the clapper/bell system or the whole belltower structure

In section 2.7 during the discussion of collaborations with artists I was disappointed to not see a more detailed discussion of this aspect of the work. As the artistic collaborations involved visual (dance) and auditory (soundscape) pieces I understand that it is difficult to convey their content in a written article. However as this collaboration is one of the unique and innovative aspects of this work I had hoped to see some more reflection on this work, maybe in the form of quotes from the performers or audience describing their emotional responses to the work (or even in the form of mood boards or wordclouds )

In section 4 (line584) the raspberryshake citizen sensors are described as using MEMS sensors. While raspberryshake systems are available as MEMS based systems their sensitivity is so low that they only work as strong motion sensors in zones of high seismicity. In the UK the raspberryshake data analysed in this paper all comes from the geophone based raspberryshake sensors which use a conventional geophone system (with a natural low frequency limit of 4.5hz which is electronically modified to give it a frequency response of 1Hz-40Hz)

Technical corrections typo line 282 "understand" should read "underside" typo line 543

"in and impermeable" should read "in an impermeable"

---

## Short Comment (SC1) · 13 Jul 2020

Hi Paul,

Many thanks for reviewing our paper and for your useful comments - particularly with regards to the artistic collaborations. We will work on incorporating all of your suggestions in the final draft of the manuscript.
* * *

---

## Referee Comment (RC2) · Susanne Maciel (Referee) · 17 Jul 2020

This paper brings a bold and lovely study of urban seismology that accounts for the case of a seismometer that monitors unscheduled bell-ringings from the Great George tower bell, in Bristol, UK. The authors explored the relationship between scales of Earth-shaking and thresholds of habituation. The authors reveal the scope that an interdisciplinary look at the vibrations of the planet can achieve, from the point of view of geophysical analysis of the seismic signal, to artistic manifestations. The authors explored from ancient works, like poems which brought interpretations for the sensations caused by earthquakes, to modern art works, even some created from the signal registered for the research.

[Figure]

**1 General comments**

Reading this work led me to two papers that I find very interesting, and I would like to share it with the authors:

a) Urban Seismology: On the origin of earth vibrations within a city,November 2017, Scientific Reports 7(1), DOI: 10.1038/s41598-017-15499-y

b) Seismometers Within Cities: A Tool to Connect Earth Sciences and Society, Front. Earth Sci., 05 February 2020 | https://doi.org/10.3389/feart.2020.00009 (Jordi Diaz , Martin Schimmel, Mario Ruiz and Ramon Carbonell)

When I first read the paper, I thought (by the abstract) that it would be a paper on urban seismology, something in the line that Díaz et al (2020) had done in their work about urban seismic noise in Barcelona. If this was the case, I would have missed a discussion on the physics that underlies the bell-ringing due to ground shaking, and a more refined exploration of different urban sources that were identified in the seismogram.

But in my opinion, the main contribution of this research is the interdisciplinary aspects of urban monitoring, which is indeed very interesting, but it is not what the abstract emphasizes. I suggest the authors to emphasize that this paper is interdisciplinary research that will investigate human connection to seismic vibrations under the point of view of music, poems, theatre and geophysics. And I agree with Paul Denton's comment that more reflection on the collaboration with artists would be very welcome.

**2 Specific comments**

**L190** It would be nice to draw an arrow or a square showing the events of the bell-ringing in Figure 4. Specially because this paper might be read by non geophysicists.

**L255** Missed reference for the Indonesian tsunami, that is present in the references list.

**L245** I suggest to specify the station name

**L255** Where is Market Rasen?

**L313** Where the authors put: "A look at the data. . ." I suggest to indicate a Figure

**Figure 5** The plot shows the whole band (0.01 to 50Hz) or you used a filter? I would like to suggest the plot of a filtered version of this Figure, for the band 15Hz to 25 Hz, in which the authors observed the Great George peals (or I would give a try to 20-30Hz band, by Figure 6 ). Maybe the other events that are visible in Figure 5 would vanish, or appear in some other frequency range? There are some peaks that seems to be regular, would they be subway noise, or something like that?

**3 Technical corrections**

Please note that I am not an English native speaker, and my English is brazilian biased. I will point out some typos that I found, and give some suggestions that I kindly ask the authors to evaluate if they are pertinent. I put in red what I think should be erased, and in blue new words or letters that should be included.

**L10,L195** I think "bell-ringing" has a hyphen

**L14** nearly continuous, or near-continuous instead of near continuous

**L19** earth-shaking events (add a hyphen)

**L20** Replace "in order to" with only "to"

**L53** the authors had already used the expression "daily basis", so I suggest changing to "ground shaking daily"

**L66** remove comma "in a different and more connected way"

**L73** "others in minutes and in millions of years"

**L77** "partly from professional activities but also from personal trajectories"

**L196,L334** I think "zoomed-in" has a hyphen

**L196** remove "clearly"

**L200** Change "there are three noteworthy features that are" to "three noteworthy features are"

**L220,L249** "high-frequency" has a hyphen

**L250** "often requires"

**L258** "The P-wave and S-wave arrivals are clearly visible at station LMK"

**L286** "there are a number of many current research endeavours to study these more exotic signals."

**L295** I think "Modern-day" seismologists has a hyphen

**L302** "signal is clearly visible above the regular activity near the tower (see Fig. 10)"

**L303** "the Greece event occurred closer to Bristol and was, therefore, more clearly visible." (add commas)

**L321** "amplified in tall buildings particularly, when they are impacted by low" (remove comma)

**L358** "This calculation shows that there is clearly a large amount of energy"

**L403** "presents these sounds via three subwoofers and..."

**L421** "media player will outlast the 21st-century"

**L535** "geothermal activity, or other catastrophes" (plural)

**L543** formations that contain gas trapped in and impermeable"

**L584** "use a MEMS (micro-electro-mechanical-systems)"

**L586** "and there are currently more that than 50 Raspberry Shake"

**L617** "where the waves travel around the inside of the Earth are is"

**L618** "Reciprocally In a reciprocal way, such realizations"

**L625** "by expanding the reach of our hearing and extending our own auditory map"

**L643** "vibrations travelling through"

**L652** "Aknowledgements"

**L669** "The numbers in the upper right corner of each trace indicates"

**L683** Spectograms

**L690** "between signal duration on bother"

**L379** (Blakeborugh, 2001) check citing format

**L436** (see also Nicols(2014)) check citing format

---

## Author Comment (AC1) · 14 Aug 2020

**Dear Dr Denton,**

**Many thanks for taking the time to review our article. The original text from your review is in black while our responses are in blue.**

General Comments This is a fascinating study which brings together seismologists, artists and researchers from the schools of English, Earth Science and Art History at the University of Bristol. This unique variety of expertise allows for the analysis of seismic signals and their perception by people in ways that have not been considered before, bringing a refreshing perspective to seismic studies from a new angle.

Specific comments: In section 2.5 in the last couple of sentences there is some confusion about the relationship between earthquake magnitude, signal amplitude and event energy release. As written it implies that the relationship between magnitude and amplitude (x10 for one unit) is similar to that for energy but then goes on to say that this is a x32 per unit magnitude relationship. This could be clarified to differentiate better between the two relationships. - - the phrasing on these sentences will be changed to make the meaning clearer to the readers.

In section 2.6 (line 351-352) it states that a Modified Mercalli intensity value of V -VI corresponds to a magnitude 5 on the richter scale. In order to avoid confusion between intensity and magnitude concepts it would be better to modify this statement along the lines of " ... corresponds to the shaking experienced close to the epicentre of an earthquake with a magnitude of 5 ... " corrected

In section 2.6 (line384) it states that the mechanism for bells ringing remains unclear. I would have liked to see a discussion at this point about the possible effects of resonance between the seismic wave frequency and the natural oscillation frequency of either the clapper/bell system or the whole belltower structure : We will work on making this discussion more robust and try to include tie-ins to the fundamental frequency of towers and possible resonance effects.

In section 2.7 during the discussion of collaborations with artists I was disappointed to not see a more detailed discussion of this aspect of the work. As the artistic collabo- rations involved visual (dance) and auditory (soundscape) pieces I understand that it is difficult to convey their content in a written article. However as this collaboration is one of the unique and innovative aspects of this work I had hoped to see some more reflection on this work, maybe in the form of quotes from the performers or audience describing their emotional responses to the work (or even in the form of mood boards or wordclouds ): Thanks for pointing this out; we have gathered some thoughts from attendees and participants of these artistic workshops and will incorporate them into the revised manuscript.

In section 4 (line584) the raspberryshake citizen sensors are described as using MEMS sensors. While raspberryshake systems are available as MEMS based systems their sensitivity is so low that they only work as strong motion sensors in zones of high seismicity. In the UK the raspberryshake data analysed in this paper all comes from the geophone based raspberryshake sensors which use a conventional geophone system (with a natural low frequency limit of 4.5hz which is electronically modified to give it a frequency response of 1Hz-40Hz) – We will change the text to reflect this variation in UK raspberry shakes.

Technical corrections typo line 282 "understand" should read "underside" typo line 543 - corrected

"in and impermeable" should read "in an impermeable" - corrected

---

## Author Comment (AC2) · 14 Aug 2020

**Dear Dr Maciel,**

**Many thanks for your thorough review of our article – I have included your original text below in black along with our responses in green.**

**1 General comments**

Reading this work led me to two papers that I find very interesting, and I would like to share it with the authors:

 a)  Urban Seismology: On the origin of earth vibrations within a city,November 2017, Scientific Reports 7(1), DOI: 10.1038/s41598-017-15499-y

I read the Urban Seismology paper and attempted some of their analysis for 2 separate weeks of data - a week in May and one in December. I do not report this analysis in the paper because the trend that we observed was somewhat different from what they reported where weekends are quieter than weekdays. As demonstrated in figures 4 to 6, weekend activity near the university appears to be higher than weekday activity. The data also showed no significant activity around New Years that could have been interpreted as fireworks. Regarding the regularity in the signal that the observed (~2 minutes on/30 seconds off) that they observed, we also don't see something similar in our dataset – for us, the nearest train station is about 1 km away from the seismic station and the track runs ~ E-W at that location. It's difficult to discern anything that coincides with the train times in our dataset.

b) Seismometers Within Cities: A Tool to Connect Earth Sciences and Society, Front. Earth Sci., 05 February 2020 | https://doi.org/10.3389/feart.2020.00009 (Jordi Diaz , Martin Schimmel, Mario Ruiz and Ramon Carbonell) – will review.

When I first read the paper, I thought (by the abstract) that it would be a paper on urban seismology, something in the line that Díaz et al (2020) had done in their work about urban seismic noise in Barcelona. If this was the case, I would have missed a discussion on the physics that underlies the bell-ringing due to ground shaking, and a more refined exploration of different urban sources that were identified in the seismogram.

But in my opinion, the main contribution of this research is the interdisciplinary aspects of urban monitoring, which is indeed very interesting, but it is not what the abstract emphasizes. I suggest the authors to emphasize that this paper is interdisciplinary research that will investigate human connection to seismic vibrations under the point of view of music, poems, theatre and geophysics. And I agree with Paul Denton's comment that more reflection on the collaboration with artists would be very welcome. We will work on incorporating your suggestions in the abstract.

**2 Specific comments**

L190 It would be nice to draw an arrow or a square showing the events of the bell- ringing in Figure 4. Specially because this paper might be read by non-geophysicists. I think the bell signals are very clear in the 10 pm spectrogram (especially in the frequency bands that I mention) and there would be too many arrows on the Charter Day chime which would make the figure too busy.

L255 Missed reference for the Indonesian tsunami, that is present in the references list. I'm not sure what you mean – Goda 2019 is cited both in the text and listed in the references

L245 I suggest to specify the station name -added
L255 Where is Market Rasen? We will include the county plus latitude/longitude for Market Rasen in the revised manuscript.

L313 Where the authors put: "A look at the data. . ." I suggest to indicate a Figure - Specific figures are referred to in the subsequent paragraph but the introductory paragraph introduces the topic so I'm not sure that referring to a specific figure here is appropriate.

Figure 5 The plot shows the whole band (0.01 to 50Hz) or you used a filter? I would like to suggest the plot of a filtered version of this Figure, for the band 15Hz to 25 Hz, in which the authors observed the Great George peals (or I would give a try to 20-30Hz band, by Figure 6 ). Maybe the other events that are visible in Figure 5 would vanish, or appear in some other frequency range? There are some peaks that seems to be regular, would they be subway noise, or something like that? - There does not appear to be a significant difference in the helicorder record filtered between 15 - 30 Hz band from the unfiltered data when looking at a full day (see the figure below which shows the same data as in figure 5b now filtered between 15-30 Hz). I really do not observe any regularity in the pattern like Diaz saw in his data set. This is probably because the closest train station is ~ 1 km away from the site and there are no underground systems in Bristol. Further, while the tower sits between two traffic lights (less than 100 m away from either light), I have not observed any true regularity to the length of the traffic lights and hence in the data. The Bristol City Council uses an automated traffic light system that manages the traffic lights in real time based on the traffic approaching junctions and crossings; as such, I believe the length of the traffic lights is variable throughout the day as traffic volume changes.

[Figure]

**3 Technical corrections**

Please note that I am not an English native speaker, and my English is brazilian biased. I will point out some typos that I found, and give some suggestions that I kindly ask the authors to evaluate if they are pertinent. I put in red what I think should be erased, and in blue new words or letters that should be included. We will incorporate the majority of the changes that you have suggested here.

L10,L195 I think "bell-ringing" has a hyphen corrected

L14 nearly continuous, or near-continuous instead of near continuous - corrected

L19 earth-shaking events (add a hyphen) corrected

L20 Replace "in order to" with only "to" corrected

L53 the authors had already used the expression "daily basis", so I suggest changing to "ground shaking daily" corrected

L66 remove comma "in a different and more connected way" corrected

L73 "others in minutes and in millions of years" corrected

L77 "partly from professional activities but also from personal trajectories" corrected

L196,L334 I think "zoomed-in" has a hyphen corrected

L196 remove "clearly" corrected

L200 Change "there are three noteworthy features that are" to "three noteworthy features are" corrected

L220,L249 "high-frequency" has a hyphen corrected

L250 "often requires" corrected

L258 "The P-wave and S-wave arrivals are clearly visible at station LMK" While we are aware that stating that something is "clear" to see is a bit of a sticky subject, we use the term clearly to signify how much stronger the signal appears on the nearby station as opposed to ours. We do go through the exercise of explaining what P and S waves are so I think that the non-geophysicist would be aware of what to look for and not have a hard time finding the signal "clear" to see

L286 "there are a number of many current research endeavours to study these more exotic signals." corrected

L295 I think "Modern-day" seismologists has a hyphen corrected

L302 "signal is clearly visible above the regular activity near the tower (see Fig. 10)" – removed

L303 "the Greece event occurred closer to Bristol and was, therefore, more clearly visible." (add commas) corrected

L321 "amplified in tall buildings particularly, when they are impacted by low" (remove comma) corrected

L358 "This calculation shows that there is clearly a large amount of energy" corrected

L403 "presents these sounds via three subwoofers and..." corrected
L421 "media player will outlast the 21st-century" corrected
L535 "geothermal activity, or other catastrophes" (plural) – corrected

L543 formations that contain gas trapped in and impermeable" corrected

L584 "use a MEMS (micro-electro-mechanical-systems)" corrected
L586 "and there are currently more that than 50 Raspberry Shake" corrected

L617 "where the waves travel around the inside of the Earth are is" corrected

L618 "Reciprocally In a reciprocal way, such realizations" corrected

L625 "by expanding the reach of our hearing and extending our own auditory map" corrected

L643 "vibrations travelling through" - corrected
L652 "Aknowledgements" -corrected
L669 "The numbers in the upper right corner of each trace indicates" - corrected

L683 Spectograms - corrected
L690 "between signal duration on bother" corrected
L379 (Blakeborugh, 2001) check citing format

 L436 (see also Nicols(2014)) check citing format